# Influence of Prevalence of Psychoactive Substance Use in Mexican Municipalities on Early Childhood Development

**DOI:** 10.3390/ijerph181910027

**Published:** 2021-09-24

**Authors:** Francisco-Javier Prado-Galbarro, Copytzy Cruz-Cruz, Jorge-Ameth Villatoro-Velázquez, Juan-Manuel Martínez-Núñez

**Affiliations:** 1Orphan Drugs Laboratory, Department of Biological Systems, Universidad Autónoma Metropolitana Campus Xochimilco, Mexico City 04960, Mexico; frjavipg@gmail.com (F.-J.P.-G.); copytzycruz@gmail.com (C.C.-C.); 2Direction of Epidemiological and Psychosocial Research, Ramón de la Fuente Muñiz National Institute of Psychiatry, Mexico City 14370, Mexico; ameth@imp.edu.mx

**Keywords:** child development, child welfare, drug use, neighborhoods, Mexico

## Abstract

Children’s early development is influenced by characteristics of the child, family, and environment, including exposure to substance abuse. The aim was to examine the association of early childhood development (ECD) with the prevalence of psychoactive substance use in Mexican municipalities. We obtained ECD data from the 2015 Survey of Boys, Girls, and Women (ENIM, for its Spanish acronym), measured with the ECD Index. The prevalence of psychoactive substance use was estimated at the municipal level, using the 2016 National Survey of Drug, Alcohol, and Tobacco Use (ENCODAT, for its Spanish acronym). Multilevel logistic models were fitted to evaluate the association between drug use and inadequacies in ECD overall and in four specific ECD domains: socio-emotional, literacy-numeric, learning, and physical. Inadequate ECD was directly associated with illegal drug use (OR = 1.10; 95% CI: 1.03, 1.17). For the specific ECD domains, inadequate socio-emotional development was directly associated with illegal drug use (OR = 1.08; 95% CI: 1.01, 1.15). These findings suggest that exposure to illegal drug use may influence ECD, and especially can lead to socio-emotional problems, although this cannot be considered the unanimous determinant of the problems presented. The implementation of evidence-based interventions to prevent drug abuse is necessary.

## 1. Introduction

Early childhood development (ECD) represents a key window of opportunity for human development, and has recently received international acknowledgement as part of the United Nations Sustainable Development Goals [1]. Children experience an extremely rapid development process, which makes them vulnerable to adverse living conditions [2]. Poor children have a lower quality of ECD than their more affluent peers, and interventions to correct these socioeconomic disparities often come too late: early life interventions are essential for individuals to attain their optimal development [3,4].

Various theoretical models have been proposed to analyze the potential influence of physical, social, and natural contexts on ECD. In 1929, Vygotsky [5] proposed a model of development that matched the basic principles of early childhood education and distinguished two kinds of development: natural and cultural. In 1994, Bronfenbrenner [6] proposed an ecological system comprising five subsystems (microsystem, mesosystem, exosystem, macrosystem, and chronosystem), each of which contributes to development. Goldfeld et al. [7] examined the influence of the neighborhood and described five domains that influence child development: physical, social, service, socioeconomic, and governance.

Evidence from neuroscience, molecular biology, and genetics has identified ECD as a critical stage defining vulnerabilities and determinants of long-term health, well-being, and opportunities [8]. There is broad scientific foundation for the idea that exposure to biological and psychosocial risks, as well as the presence of vulnerability factors (including poverty, marginalization, and use of psychoactive substances), could negatively affect the physical, cognitive, and socio-emotional dimensions of child development [4,9].

Existing literature on the topic suggests that the neighborhood environment has been identified as key for helping families to cope with adversities and threats that could affect child development [1,3]. Exposure to drugs is one of the many disadvantageous neighborhood influences on child development. There is evidence that children living in negative environments have greater exposure to substances and more opportunities to use them [10]. The use of psychoactive substances has repercussions on the user’s family and social environment [11]. Serino et al. [12] found that mothers who used methadone and cocaine presented with more severe prenatal drug use and psychosocial risk factors than women who used primarily marijuana. NIDA-funded studies showed that children who were prenatally exposed to illegal drugs may be at risk of later behavioral and learning difficulties [13,14,15,16].

All these investigations coincide in pointing out the impact of parental drug use on child development. However, to the best of our knowledge, no prior research has examined how the frequency of use of psychotropic substances in a municipality in which the child is growing up impacts ECD. This study has compiled information from various sources, in order to explore how the prevalence of psychoactive substance use in municipalities impacts overall ECD and its specific domains in Mexico. The main hypothesis was that children with exposure to non-prescription use of medical drugs and illegal drug use could present language, motor, and cognitive deficits during growth. On the other hand, children living in municipalities with the highest number of homicides in Mexico would have worse ECD outcomes.

## 2. Materials and Methods

### 2.1. Study Design and Participants

This study was a secondary analysis of data collected from two surveys implemented in México: the 2015 National Survey of Boys, Girls, and Women (Encuesta Nacional de los Niños, Niñas y Mujeres, ENIM) and the 2016 National Survey of Drug, Alcohol, and Tobacco Use (Encuesta Nacional de Consumo de Drogas, Alcohol y Tabaco, ENCODAT). Both surveys have a probabilistic, multistage, stratified, and clustered sample design, and are representative at the national, regional, and urban/rural levels. The present study used a nationally representative sample of children aged 36–59 months from the ENIM, and we linked individual-level data with drug use data at the municipal level from the ENCODAT and other sources of neighborhood-level data in Mexico.

The ENIM oversampled children under five years old and in rural areas to estimate indicators for children and women in 2015. It employed four questionnaires: (1) a household questionnaire, (2) a questionnaire for women aged 15–49 years, (3) a questionnaire about children and adolescents aged 5–17 years, answered by their mothers, and (4) a questionnaire about children under 5 years of age, answered by their mothers [17]. We selected 1818 children for our analysis, with information extracted from the questionnaires for the household, for women, and for children under five. The latter questionnaire included data modules for birth registration, ECD, lactation and food intake, vaccination, disease care, functioning and disability (at age 2–4 years), anthropometry, and hemoglobin.

The ENCODAT sample consisted of 56,877 people who answered a standardized audio computer-assisted self-interviewing (ACASI) questionnaire that collected information about the use of tobacco, alcohol, and medical and illegal drugs in Mexico [18]. There were two questionnaires: one about the household and the other about the individual. The first was applied to the head of household, homemaker, or another member of the family aged 18 or over who had no difficulty answering and who knew the characteristics of the dwelling and its usual residents. The second was applied to a randomly selected individual, aged 12–17 or 18–65 at the time of the visit.

We used a harmonized dataset of individual and municipal-level data for 145 municipalities. Based on the child’s place of residence, we linked individual-level with municipal-level data using unique municipality codes. Municipalities are second-level administrative divisions (states being the first). They have legislative and executive authority and are responsible for the provision of basic public services for their population.

### 2.2. Study Measurements

#### 2.2.1. Psychoactive Substance Use

The prevalence of non-prescription use of medical drugs (opiates, tranquilizers, sedatives, and amphetamines) and of illegal drug use (marijuana, cocaine, crack, hallucinogens, inhalants, heroin, and methamphetamines) in the 12 months prior to the survey was calculated using ENCODAT data. Then the respondents were asked, “Have you ever, even once, used marijuana, cocaine, crack, hallucinogens, inhalants, heroin or methamphetamines in the past 12 months?” and “Have you ever, even once, used opiates, tranquilizers, sedatives or amphetamines in the past 12 months?”. Therefore, we considered two binary variables: non-prescription use of medical drugs (Yes/No) and illegal drug use (Yes/No).

Small area estimation (SAE) was used with data from the ENCODAT to estimate psychoactive substance use by municipality. Two SAE methodologies were used: direct and indirect [19]. Direct estimations are based on survey samples, and the Horvitz-Thompson estimator is generally used to produce official statistics [20]. However, there are some problems inherent in survey samples, as they produce SAE without sufficient statistical precision. A good alternative is the use of indirect SAE, which can be calculated using a statistical model. Auxiliary information for small areas was aggregated to increase survey sampling precision. We used area-level models described by Fay-Herriot [21], which link the estimated average of the variable in an area *d* with auxiliary information, and explored various datasets as sources of that information. The auxiliary variables, considered by municipality, were crime rate [22], social backwardness index [23], and population density [24]. The general form of the area-level models, assuming a logit link function, was:(1)$$\hat{\bar{Y_{d}}}=z_{d}^{t}\beta +u_{d}+e_{d}$$
where $\hat{\bar{Y_{d}}}$ is the estimated average of the variable of interest in the area (in our case, the prevalence of non-prescription use of medical drugs and of illegal drug use), $z_{d}^{t}$ is the transpose of the vector of predictors computed at the *d*th municipality (auxiliary municipal information), u_d_ ~ N(0, $\sigma_{v}^{2}$) are independent and identically distributed area-specific random effects, and e_d_ ~ N(0, Ʃ_e_) represent sampling errors within the *d*th municipality with Ʃ_e_ = diag(D_1_, D_2_, …, D_d_).

We finally selected the Fay-Herriot estimator because it improved direct estimator in terms of accuracy. We used the R ecosystem to implement the SAE methodology, and a Fay-Herriot model for estimated proportions was adjusted with the function mseFH from the SAE library [25].

#### 2.2.2. Study Outcome: Early Childhood Development

We obtained individual data from ENIM. Children were considered to have adequate ECD if they met the age-expected development targets in at least three of the following four domains: socio-emotional, literacy-numeracy, learning, and physical. These domains were measured using a 10-item module developed and validated by UNICEF [26]. Each domain was used as an independent outcome (binary variables) and in combination, by creating an ECD index. For literacy-numeracy (3 items), development was considered to be adequate if a child could do at least two of the following: identify or name at least ten letters of the alphabet, read at least four common simple words, or recognize all of the numerals from 1 to 10. Adequate socio-emotional development (2 items) was obtained with at least two affirmative answers to the child gets along well with other children, does not kick, bite, or hit adults or other children, and does not get distracted easily. Physical development (2 items) was considered adequate if a child could pick up small objects (a rock or a stick) with two fingers from the ground, and/or if the mother/primary caregiver indicated that the child was well enough to play. Learning (2 items) was considered adequate if a child could follow simple instructions on how to do things, and/or if he/she could independently follow an instruction given by an adult. The ECD index was then defined as the percentage of children aged 36–59 months who were developmentally on track in at least three of these four domains, and the children were categorized as having adequate or inadequate ECD [26].

#### 2.2.3. Child-Level Covariates

Children’s characteristics included as covariates were sex, age in months, presence of functional difficulties in at least one domain (seeing, hearing, walking, fine motor coordination, understanding, being understood, learning things, playing, and, where applicable, controlling behavior), mother’s age and education, ethnicity (indigenous/not indigenous), and wealth quintiles for households. Children were considered to have a functional difficulty if they presented severe difficulties in at least one of the following functional areas: seeing, hearing, walking, fine motor, communication (understanding or being understood), learning, playing, or behavior. Mother’s education was the maximum level of school attendance achieved by the mother (none, primary, secondary, high school, or college and higher). To adjust for household wealth, a composite indicator was calculated in the ENIM using principal component analysis, including floor material, roof and exterior walls, fuel used for cooking, electricity, radio, television, landline, refrigerator, DVD, microwave, computer, cable TV, internet, washing machine, gas stove, water heater, water tank, clock, cellphone, bicycle, motorcycle or scooter, automobile, motorboat, laptop, bank account, source and availability of drinking water, type of sanitary installation, and availability of materials to wash hands at home. Households were classified into five wealth categories, from the poorest to the wealthiest quintile [17].

#### 2.2.4. Area-Level Covariates

A number of municipality-level covariates were included with a possible relation to psychoactive substance use and ECD. These included population density [24], marginalization index [27], and number of reported homicides [28] (using INEGI homicide data from death certificates, which tends to be more precise than Mexican police records).

### 2.3. Statistical Analysis

Individual and municipal characteristics were analyzed by ECD status, using means and standard deviations for continuous variables, and percentages and 95% confidence intervals for categorical variables. Bivariate analyses were conducted to determine the association between individual or municipal variables and inadequate ECD and each of its domains. A final model for each outcome was fitted including psychoactive substance exposure variables, and all significant environment variables from bivariate models, plus individual-level covariates (sex, age in months, functional disability, mother’s age, mother’s education, household wealth quintile, ethnicity, and domestic violence). The association between ECD and psychoactive substance use was analyzed by fitting two-level logistic regression models (first level children and second level municipalities) and random intercepts for neighborhood to account for correlations in ECD response among children living within the same municipality, as follows:(2)$$y_{ij}\;\sim \;\mathrm{Binomial}(1,\;\pi_{ij})$$
(3)$$\log\left(\frac{\pi_{ij}}{1-\pi_{ij}}\right)=\beta_{0}+\sum_{k=1}^{n}\beta_{k}X_{ki}+\sum_{l=1}^{m}\beta_{l}Z_{lj}+u_{j}$$
where *π_ij_* is the probability of inadequate ECD for child *i* in municipality *j*, *X_i_* is the set of explanatory variables at the individual level (level 1), *Z_j_* is the set of explanatory variables defined for the municipalities (level 2), and *u_j_* the residuals of level 2, for which it is assumed that they are independent and follow a normal distribution with mean 0 and variance $\sigma_{u}^{2}$.

A null model was fitted to estimate the median odds ratio (*MOR*), which is a measure of the variation of ECD across different municipalities that is not explained by the modeled risk factors, obtained with the following formula:(4)$$MOR=\exp\left[\sqrt{\left(2*V_{A}\right)}*\;0.6745\right]\;\approx \exp\left(0.95\;\sqrt{V_{A}}\right)\;$$
where *V_A_* is the municipality-level variance. If the *MOR* is 1, there is no variation between municipalities; a larger *MOR* indicates variation.

The analysis accounted for complex survey design and all estimates presented here were weighted. Data analysis was performed using Stata 14.0 (StataCorp, College Station, TX, USA).

### 2.4. Ethics Approval and Informed Consent

The ENIM and ENCODAT were conducted according to the guidelines laid down in the Helsinki Declaration, and all procedures involving human subjects were approved by the Ethics Committee of the Mexican National Institute of Public Health (Project No. 1033, Approval No. 1108). Written informed consent was obtained from all participants. All analyses in the present study were performed with de-identified secondary data.

## 3. Results

Table 1 presents the percentage of children with inadequate overall ECD. A total of 17.3% of the children showed inadequate ECD, with boys showing a higher percentage of inadequate ECD than girls (22.3% and 12.5%, respectively). Children and mothers in the inadequate ECD group tended to be younger than in the adequate group. A higher proportion of children with functional difficulties and disabilities had inadequate ECD than those without these problems (31% and 16.5%, respectively). The proportion of children with inadequate ECD was significantly higher for mothers with primary education or less (21%) and lower for those with college (8.2%) than for those with a middle or high school education (20.7%% and 14.6%, respectively). Children who lived in the poorest households (23.9%) and in rural areas (20.2%) had a higher prevalence of inadequate ECD than those from the wealthiest households (9.5%) and urban areas (16.2%). Finally, a higher prevalence of inadequate ECD was observed in children who were exposed to domestic violence than in those who did not (18.7%% and 12.8%, respectively). Details of ECD for each domain are shown in Appendix A.

Table 2 shows the prevalences of psychoactive substance use and municipality characteristics for children with inadequate and adequate ECD. Children with inadequate ECD lived in municipalities with a greater prevalence of illegal drug use than children with adequate ECD (3.7% vs. 3.3%). Children with inadequate ECD lived in municipalities with lower population density than those with adequate ECD (an average of 2267 vs. 3157 inhabitants per km^2^). Children with inadequate ECD lived in areas with higher prevalence of non-prescription use of medical drugs and of use of drugs with and without dependence, and in areas with a higher marginalization index and number of homicides, but these associations were not statistically significant. Details of municipality characteristics according to children’s ECD for each domain are shown in Appendix A.

### 3.1. Association between Psychoactive Substance Use and Inadequate ECD 

Table 3 presents the odds ratio of inadequate ECD and psychoactive substance use adjusted by municipality-level and individual characteristics. From the null model, it was estimated that the median odds of inadequate ECD in a municipality with a high prevalence of inadequate ECD would be double that of one with a low prevalence (MOR = 2.08; 95% CI: 1.60, 2.57), indicating a substantial heterogeneity across municipalities. With the inclusion of individual and municipality-level variables in the analysis, the MOR decreased to 1.98 (95% CI: 1.55–2.42). Illegal drug use was associated with a 10% increase in inadequate ECD (OR = 1.10; 95% CI: 1.03, 1.17). A one standard deviation increase in homicides was associated with a 21% increase in the odds of inadequate ECD (OR = 1.21; 95% CI: 1.07, 1.37). Girls had a lower risk of inadequate ECD than boys (OR = 0.53; 95% CI: 0.35, 0.80). A one-unit (one-month) increase in children’s age was associated with an 8% decrease in the odds of inadequate ECD (OR = 0.92; 95% CI: 0.90, 0.95). Children with functional difficulties had 1% greater odds of presenting inadequate ECD (OR = 1.01, 95% CI: 1.01, 1.02) than those without such difficulties. Children living in poor and medium-wealth households had lower odds of inadequate ECD than those in very poor households (OR = 0.44; 95% CI: 0.24, 0.81; OR = 0.52; 95% CI: 0.27, 0.99, respectively). Children exposed to domestic violence in their household had 65% higher odds of presenting inadequate ECD (OR = 1.65; 95% CI: 1.05, 2.60) compared to those without domestic violence.

### 3.2. Association between Psychoactive Substance Use and Socio-Emotional Development

Illegal drug use and number of homicides were directly associated with an increase in inadequate ECD (OR = 1.08; 95% CI: 1.01, 1.15; OR = 1.24; 95% CI: 1.02, 1.52, respectively). Girls had a lower risk of inadequate socio-emotional development than boys (OR = 0.60; 95% CI: 0.42, 0.87). A one-unit (one-month) increase in children’s age was associated with a 6% decrease in the odds of inadequate socio-emotional development (OR = 0.94; 95% CI: 0.92, 0.97). Children with functional difficulties had 1% greater odds of presenting inadequate socio-emotional development (OR = 1.01, 95% CI: 1.001, 1.02) than those without such difficulties. Children living in poor, middle, and very wealthy households had lower odds of presenting inadequate socio-emotional development than those living in very poor households (OR = 0.54; 95% CI: 0.30, 0.99; OR = 0.447; 95% CI: 0.244, 0.818; and OR = 0.46; 95% CI: 0.22, 0.97 respectively). Children exposed to domestic violence in their household had 68% greater odds of inadequate socio-emotional development (OR = 1.68, 95% CI: 1.09, 2.60) than those without exposure to domestic violence (Table 4).

### 3.3. Factors Associated with Development of Literacy and Numeracy

A one-unit (one-month) increase in children’s age was associated with 9% lower odds of inadequate ECD (OR = 0.91; 95% CI: 0.88, 0.93). Children of mothers with high school had 47% lower odds of presenting inadequate development of literacy and numeracy (OR = 0.53, 95% CI: 0.29, 0.97) than those who mothers had primary education or less. Children living in poor, middle, wealthy, and very wealthy households had lower odds of inadequate ECD than those in very poor households (OR = 0.28; 95% CI: 0.14, 0.60; OR = 0.29; 95% CI: 0.14, 0.60; OR = 0.25; 95% CI: 0.11, 0.57 and OR = 0.27; 95% CI: 0.11, 0.64, respectively) (Table 4).

### 3.4. Factors Associated with Learning

A one-unit (one-month) increase in children’s age was associated with 15% lower odds of inadequate learning (OR = 0.85; 95% CI: 0.80, 0.90). Girls had a lower risk of inadequate learning than boys (OR = 0.24; 95% CI: 0.12, 0.48). Children with functional difficulties had greater odds of presenting inadequate learning (OR = 1.02; 95% CI: 1.01, 1.03). Those with mothers with a high educational level and those living in poor households had lower odds of presenting inadequate learning than those with mothers with less education or who lived in very poor households (OR = 0.334, 95% CI: 0.112, 0.993; OR = 0.381, 95% CI: 0.154, 0.941; respectively) (Table 4).

### 3.5. Factors Associated with Physical Development

Girls had a lower risk of inadequate physical development than boys (OR = 0.47; 95% CI: 0.23, 0.96). Children with functional difficulties had 2% greater odds of inadequate physical development (OR = 1.023, 95% CI: 1.01, 1.04) than those without such difficulties. Those living in poor households had lower odds of presenting inadequate physical development (OR = 0.27, 95% CI: 0.08, 0.87) than those in very poor households (Table 4).

## 4. Discussion

This study evaluated the association between the prevalence of psychoactive substance use and inadequate ECD, both overall and in four ECD domains. We found that greater illegal drug use in Mexican municipalities was associated with greater odds of inadequate ECD. Illegal drug use was also associated with higher odds of inadequate development in the socio-emotional domain, independent of household, maternal, and individual characteristics. 

These findings suggest that neighborhood disadvantage may affect children’s development. In this study, we did not have information on whether the children’s parents used illegal drugs, but we found a significant association between illegal drug use in the neighborhood and inadequate ECD, which can lead to behavioral problems in children by the indirect effects of the consumption of these substances generates, such as abandonment, lack of care, and child maltreatment. Micol Parolin et al. found that parental substance use was a major risk factor for child development [29]. Furthermore, most current academic research has studied effects of prenatal substance exposure [30,31,32]. Emily J Ross et al. [33] found that fetal exposure to drugs could cause brain damage in the infant and balance problems. Other preliminary studies pointed out that maternal cocaine abuse during pregnancy had a negative impact on mothers, children, families and society. These children presented various neurobehavioral problems, such as anxiety, depression, stress, among others [34,35,36].

Evidence consistently shows an important interaction between drug use and crime [37,38]. We found that a greater number of homicides was associated with higher odds of inadequate ECD and socio-emotional development on the municipal level in Mexico. When children are exposed to a violent crime, they may develop different regulatory processes, such as that they may prefer to stay at home, and they may have trouble sleeping and difficulty with concentrating [39].

Our findings are consistent with the results of the U.S. National Institute on Drug Abuse (NIDA) regarding the effects of drug abuse on education and the family, and its contribution to violence, crime, and financial and housing problems [40]. Although situations of vulnerability could occur at different points along the life course, extensive research in psychology and human development has shown that features of the social and physical environment and life events experienced in childhood contributed to children’s physical, socio-emotional, and cognitive development [41,42,43]. 

There are gender disparities in overall ECD, with girls demonstrating better outcomes than boys. This pattern is also evident in the domains of socio-emotional development and learning. Studies have found that girls have an advantage in socio-emotional development [44] and boys have an advantage in physical development [45]. Older children are also more likely to have adequate ECD. A study by psychologists Betty Hart and Todd Risley showed that children from lower-income families heard a staggering 30 million fewer words than children from higher-income families during the first three years of life, and that the difference increased with age [46]. Mothers’ education is an important facilitating factor for child development [47]. Our study found that children of mothers with lower educational levels presented greater odds of inadequate ECD. Various studies suggest that if mothers had at least a middle-school education, the odds of inadequate ECD were lower [8,48], especially in the developmental domain of literacy and numeracy. 

Our study also indicates that children with functional difficulties and disabilities had a greater risk of inadequate ECD. Disability experienced early in life may mean that children were less likely to attend school or receive health care, and that they were more vulnerable to poverty [49,50]. Children with disabilities were also more likely to be victims of violence [51,52]. Our results are in agreement with those of published studies that examined domestic violence in early childhood, which could greatly affect the lives of children and impede their development [53,54]. 

Our study has several limitations. First, the ENCODAT data are not representative at the municipality level. SAE methodology is thus commonly used by statistical agencies in various countries. Second, although the differences between the estimates obtained by direct and indirect methods were minimal, the estimation method may improve as additional information becomes available to improve the accuracy of estimates. Finally, because the data are cross-sectional, we cannot establish causal relationships; however, the results are consistent with previous studies. Despite these limitations, the ENCODAT is Mexico’s largest survey regarding addiction, and the ENIM is its first survey on ECD. This study served as an exploratory effort to identify potential associations between drug use and ECD, as well as other contextual factors. The associations found in this study and the potential mechanisms linking contextual variables with ECD will need to be confirmed in longitudinal studies.

## 5. Conclusions

To our knowledge, this research presents the first empirical evidence in Mexico of a link between psychoactive substance use and inadequate ECD. We provide evidence that exposure to illegal drugs is pervasive in the lives of children. However, although illegal use drug plays a role as an important risk factor for child development, this cannot be considered the unanimous determinant of the problems presented because other environmental factors that influence child development. These findings suggest that interventions and the implementation of public policy to prevent the use of psychoactive substances may benefit children’s development, with significant benefits for social environment, and they should be an ongoing priority in the public health field. Child development is one of the United Nations Sustainable Development Goals [55], and governments should consider interventions to improve municipal conditions and to promote early childhood development, implementing new programs for prevention and treatment for drug use and disorders in the municipalities.

## Figures and Tables

**Table 1 ijerph-18-10027-t001:** Individual Characteristics of Children with Inadequate and Adequate Early Childhood Development.

Individual Characteristics	Inadequate ECD	Adequate ECD	*p*-Value
Total children	378	1467	
Functional difficulties, %			
No	16.5 (13.16–20.49)	83.5 (79.51–86.84)	0.036 *
Yes	31.99 (17.46–51.13)	68.01 (48.87–82.54)	
Mother’s education, %			
Primary school or less	20.94 (15.09–28.29)	79.06 (71.71–84.91)	0.036 *
Middle school	20.65 (15.71–26.64)	79.35 (73.36–84.29)	
High school	14.55 (11.31–18.53)	85.45 (81.47–88.69)	
College	8.19 (3.36–18.63)	91.81 (81.37–96.64)	
Wealth quintiles for households, %			
Very poor	23.94 (16.04–34.14)	76.06 (65.86–83.96)	0.184
Poor	15.36 (11.49–20.23)	84.64 (79.77–88.51)	
Middle	20.18 (11.91–32.09)	79.82 (67.91–88.09)	
Wealthy	17.76 (12.43–24.74)	82.24 (75.26–87.57)	
Very wealthy	9.54 (4.32–19.77)	90.46 (80.23–95.68)	
Sex, %			
Male	22.27 (19.16–25.72)	77.73 (77.73–77.73)	0.007 *
Female	12.54 (8.23–18.65)	87.46 (81.35–91.77)	
Indigenous, %			
No	16.95 (12.84–22.05)	83.05 (77.95–87.16)	0.873
Yes	16.4 (12.06–21.91)	83.6 (78.09–87.94)	
Area, %			
Urban	16.18 (12.53–20.63)	83.82 (79.37–87.47)	0.229
Rural	20.24 (15.35–26.22)	79.76 (73.78–84.65)	
Domestic violence, %			
No	12.81 (9.30–17.40)	87.19 (82.60–90.70)	0.120
Yes	18.66 (13.62–25.02)	81.34 (74.98–86.38)	
Mother’s age	28.13 (7.49)	29.92 (6.36)	0.023 *
Child’s age (months)	45.62 (8.19)	49.51 (6.62)	<0.001 *

* Significant value (*p* < 0.05).

**Table 2 ijerph-18-10027-t002:** Prevalence of psychoactive substance use and environmental characteristics of the municipalities where children with inadequate or adequate early childhood development live.

Variable, Mean (*SD*)	Inadequate ECD	Adequate ECD	*p*-Value
% Using illegal drugs	3.74 (2.60)	3.28 (2.09)	0.043 *
% Using non-prescription medical drugs	0.61 (0.95)	0.52 (1.07)	0.416
Marginalization index	−1.37 (0.51)	−1.39 (0.52)	0.548
Number of homicides	147.58 (206.22)	121.44 (153.37)	0.081
Population density by municipality (inhabitants/km^2^)	2266.89 (4334.10)	3156.71 (4664.98)	0.024 *
% Drug-dependent users	0.87 (1.19)	0.75 (0.97)	0.128
% Users without drug dependence	11.61 (4.39)	11.39 (4.46)	0.607
% Non-users with exposure to drugs	18.3 (6.21)	18.72 (5.95)	0.516
% Non-users with no drug exposure	69.22 (7.11)	69.14 (6.71)	0.910

* Significant value (*p* < 0.05).

**Table 3 ijerph-18-10027-t003:** Adjusted Associations Between Overall Inadequate Early Childhood Development and Psychoactive Substance Use.

Predictor		Early Childhood Development **	
		OR	95% CI
Use illegal drugs		1.102 *	1.033–1.176
Use medical drugs without prescription		1.001	0.840–1.193
Marginalization index		0.843	0.604–1.177
Number of homicides ***		1.209 *	1.070–1.367
Population density by municipality ****		0.956	0.909–1.004
Functional difficulties	No	1 (Ref.)	
	Yes	1.012 *	1.004–1.020
Mother’s education	Primary or less	1 (Ref.)	
	Middle school	1.191	0.693–2.047
	High school	0.653	0.371–1.147
	College	0.461	0.209–1.015
Household wealth quintile	Very poor	1 (Ref.)	
	Poor	0.441 *	0.241–0.807
	Middle	0.518 *	0.271–0.989
	Wealthy	0.644	0.284–1.459
	Very wealthy	0.563	0.248–1.277
Mother’s age		0.978	0.945–1.012
Sex	Male	1 (Ref.)	
	Female	0.530 *	0.350–0.803
Child’s age (months)		0.922 *	0.896–0.949
Indigenous	No	1 (Ref.)	
	Yes	1.304	0.846–2.011
Area	Urban	1 (Ref.)	
	Rural	1.139	0.603–2.152
Domestic violence	No	1 (Ref.)	
	Yes	1.654 *	1.053–2.599
MOR			
Null model		2.082	1.598–2.567
Adjusted model		1.984	1.553–2.415

* Significant value (*p* < 0.05). ** All models were adjusted by the set of individual variables for each child: sex, age in months, functional disability, mother’s age, mother’s education, wealth quintiles for urban households, indigenous/not indigenous, rural/urban, and domestic violence. *** Re-scaled using *z*-scores, so that a one-unit change represents a one standard deviation change in municipal services of municipalities. **** Per 1000 inhabitants.

**Table 4 ijerph-18-10027-t004:** Adjusted Associations Between Psychoactive Substance Use and Inadequate Early Childhood Development, by Domain.

Predictors		Socio-Emotional **		Literacy-Numeracy **		Learning **		Physical **	
		OR	95% CI	OR	95% CI	OR	95% CI	OR	95% CI
Use illegal drugs		1.079 *	1.013–1.149	0.994	0.917–1.077	1.110	0.988–1.247	0.975	0.860–1.104
Use medical drugs without prescription		1.030	0.858–1.235	0.899	0.784–1.030	1.028	0.768–1.374	0.878	0.640–1.203
Marginalization index		0.908	0.638–1.292	0.817	0.531–1.259	1.051	0.553–1.994	1.326	0.785–2.237
Number of homicides ***		1.244 *	1.018–1.520	0.952	0.785–1.155	1.122	0.878–1.434	1.076	0.887–1.304
Population density by municipality ****		0.952	0.903–1.003	0.952	0.894–1.015	1.033	0.970–1.099	1.013	0.947–1.084
Functional difficulties	No	1 (Ref.)		1 (Ref.)		1 (Ref.)		1 (Ref.)	
	Yes	1.010 *	1.001–1.019	1.012	0.999–1.026	1.020 *	1.007–1.033	1.023 *	1.008–1.038
Mother’s education	Primary or less	1 (Ref.)		1 (Ref.)		1 (Ref.)		1 (Ref.)	
	Middle school	1.430	0.836–2.448	0.836	0.458–1.526	0.647	0.284–1.471	1.254	0.457–3.437
	High school	0.925	0.522–1.639	0.531 *	0.290–0.973	0.334 *	0.112–0.993	0.650	0.209–2.024
	College	0.654	0.309–1.381	0.593	0.275–1.281	0.194 *	0.0404–0.929	0.487	0.110–2.166
Household wealth quintile	Very poor	1 (Ref.)		1 (Ref.)		1 (Ref.)		1 (Ref.)	
	Poor	0.544 *	0.297–0.995	0.284 *	0.135–0.597	0.381 *	0.154–0.941	0.266 *	0.081–0.872
	Middle	0.447 *	0.244–0.818	0.285 *	0.136–0.595	0.488	0.163–1.463	1.075	0.445–2.596
	Wealthy	0.628	0.293–1.346	0.254 *	0.114–0.567	0.614	0.171–2.200	1.051	0.331–3.337
	Very wealthy	0.456 *	0.215–0.969	0.271 *	0.114–0.642	0.755	0.192–2.977	1.165	0.273–4.976
Mother’s age		0.977	0.948–1.006	1.014	0.983–1.046	1.012	0.946–1.082	1.003	0.938–1.073
Sex	Male	1 (Ref.)		1 (Ref.)		1 (Ref.)		1 (Ref.)	
	Female	0.604 *	0.419–0.870	0.799	0.578–1.105	0.241 *	0.121–0.477	0.466 *	0.226–0.964
Child’s age (months)		0.940 *	0.915–0.965	0.907 *	0.880–0.934	0.848 *	0.796–0.904	0.750*	0.650–0.867
Indigenous	No	1 (Ref.)		1 (Ref.)		1 (Ref.)		1 (Ref.)	
	Yes	1.401	0.916–2.141	1.289	0.832–1.996	1.074	0.433–2.664	0.868	0.335–2.246
Area	Urban	1 (Ref.)		1 (Ref.)		1 (Ref.)		1 (Ref.)	
	Rural	1.066	0.613–1.856	0.853	0.449–1.622	0.771	0.355–1.675	0.660	0.251–1.736
Domestic violence	No	1 (Ref.)		1 (Ref.)		1 (Ref.)		1 (Ref.)	
	Yes	1.682 *	1.091–2.595	1.278	0.862–1.895	1.446	0.630–3.318	1.327	0.648–2.719
MOR									
Null model		2.234	1.733–2.735	2.526	1.975–3.076	2.138	1.250–3.026	1.449	0.681–2.217
Adjusted model		2.064	1.681–2.448	2.608	1.988–3.228	1.065	−5.614–7.745	1.001	0.999–1.003

* Significant value (*p* < 0.05). ** All models were adjusted by the set of individual variables for each child: sex, age in months, functional disability, mother’s education, mother’s age, wealth quintiles for urban households, indigenous/not indigenous, urban/rural, and domestic violence. *** Re-scaled using *z*-scores, so that a one-unit change represents a one standard deviation change in municipal services. **** Per 1000 inhabitants.

## Data Availability

The datasets used and/or analyzed during the current study are publically available for download at: https://mics.unicef.org/surveys (accessed on 31 August 2021) (ENIM 2015), https://encuestas.insp.mx/ena/encodat2017.php (accessed on 31 August 2021) (ENCODAT 2016–2017).

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
