# Peer review of "Influence of Prevalence of Psychoactive Substance Use in Mexican Municipalities on Early Childhood Development"

_ijerph, 2021, doi:10.3390/ijerph181910027_

Round 1
Reviewer 1 Report
An interesting and very socially relevant topic. The study conducted a secondary analysis of two studies conducted earlier.
The results of the study are not revealing, but they confirm trends known from other studies. In addition, the research group is impressive, so this study should be appreciated.
If this is indeed the first such study in Mexico, perhaps the Authors could try to give in their conclusions some specific social policy interactions that should be implemented to improve the situation of children. If this section were expanded, the article would provide important guidance to policy makers and thus be highly useful.
Additional comments:
Correct punctuation errors, including line 61, 418.
Bring the References section (in its entirety) in line with Journal requirements.
Also:
Correct notation of reference 2: There should be:
National Research Council (US) and Institute of Medicine (US) Committee on Integrating the Science of Early Childhood Development, Shonkoff, J. P., & Phillips, D. A. (Eds.). (2000). From Neurons to Neighborhoods: The Science of Early Childhood Development. National Academies Press (US).
(adapt to Journal requirements).
Reference 3: incomplete.
Check the entire References section carefully (generally chaos in this section).
Author Response
Comments and Suggestions for Authors
An interesting and very socially relevant topic. The study conducted a secondary analysis of two studies conducted earlier. The results of the study are not revealing, but they confirm trends known from other studies. In addition, the research group is impressive, so this study should be appreciated.
R1-1. Conclusions:
If this is indeed the first such study in Mexico, perhaps the Authors could try to give in their conclusions some specific social policy interactions that should be implemented to improve the situation of children. If this section were expanded, the article would provide important guidance to policy makers and thus be highly useful.
AUTHOR RESPONSE: Thanks, we modified that paragraph as follows:
To our knowledge, this research presents the first empirical evidence in Mexico of a link between psychoactive substance use and inadequate ECD. We provide evidence that exposure to illegal drugs are pervasive in the lives of children. However, although illegal use drug plays a role as an important risk factor for child development, this cannot be considered the unanimous determinant of the problems presented because other environmental factors that influence child development. These findings suggest that interventions and the implementation of public policy to prevent the use of psychoactive substances may benefit children's development, with significant benefits for social environment, and they should be an ongoing priority in the public health field. Child development is one of the United Nations Sustainable Development Goals [55], and governments should consider interventions to improve municipal conditions and to promote early childhood development, implementing new programs for prevention and treatment for drug use and disorders in the municipalities.
R1-2. Additional comments:
Correct punctuation errors, including line 61, 418.
AUTHOR RESPONSE: Thank you, we corrected punctuation errors.
Bring the References section (in its entirety) in line with Journal requirements.
Also:
Correct notation of reference 2: There should be:
National Research Council (US) and Institute of Medicine (US) Committee on Integrating the Science of Early Childhood Development, Shonkoff, J. P., & Phillips, D. A. (Eds.). (2000). From Neurons to Neighborhoods: The Science of Early Childhood Development. National Academies Press (US).
(adapt to Journal requirements).
Reference 3: incomplete.
Check the entire References section carefully (generally chaos in this section).
AUTHOR RESPONSE: We apologize for the confusion. The references have been fixed.

Reviewer 2 Report
Many thanks for giving me the opportunity to review the revised manuscript entitled ”Association Between Psychoactive Substance Use and Early Childhood Development in Mexico “.
This study is focussed on environmental factors of early child development (ECD), therefore, its content is appropriate for the special issue "Maternal, Newborn, Child and Family Health: Past, Present, and Future" of the IJERPH.
Reading the submitted manuscript, several questions arose and drawbacks were noticed, which I recommend to fix.
- Overall: In the title, abstract and article body text the term "psychoactive substance use" must be defined in such a way as to make it clear that it is not the substance use of the individual but the frequency of use of psychotropic substances in a municipality in which the child is growing up.
- Abstract: Abstract should be revised and make it more appealing. Include conclusions. They must be fit to the aim of the study. There are logical errors (see lines 22-23). Why provide data on homicide if this is not the aim of the study?
- Introduction: Please clarify the rationale of the study, provide examples of studies by other authors that meet the stated aim. A reference is needed for the NIDA study (lines 59-61).
- Material and Methods: The assessment of substance use data is unclear (lines 114-119). Please describe this in more detail and provide references. How were the first and second level data combined? How many municipalities were included in the analysis?
- Results: For OR, it is enough to use two decimals but this is not a principal requirement.
- Discussion: Please discuss the main results of the study "the association between drug use and inadequate early childhood development (ECD), both overall and in four ECD domains." Is this a direct or indirect association? How can you explain the nature of this association? Compare with studies by other authors that have investigated such an association.
- Conclusions: There are presented only implications of the study but there are any conclusions generated in the present analyses.
- References: References must be presented accordind to the requirements of the journal.
- English spell revision is necessary.
Thank you for considering my opinion. I encourage authors to keep on working to improve the manuscript.
Author Response
Comments and Suggestions for Authors
Many thanks for giving me the opportunity to review the revised manuscript entitled “Association Between Psychoactive Substance Use and Early Childhood Development in Mexico”.
This study is focussed on environmental factors of early child development (ECD), therefore, its content is appropriate for the special issue "Maternal, Newborn, Child and Family Health: Past, Present, and Future" of the IJERPH.
Reading the submitted manuscript, several questions arose and drawbacks were noticed, which I recommend to fix.
- Overall: In the title, abstract and article body text the term "psychoactive substance use" must be defined in such a way as to make it clear that it is not the substance use of the individual but the frequency of use of psychotropic substances in a municipality in which the child is growing up.
AUTHOR RESPONSE: Thank you, we have made changes throughout the text to clarify that the term “psychoactive substance use” refers to the prevalence of use of psychotropic substances in a municipality in which the child is growing up. Also, we modified the title in order to address your comment, now it reads as follows:
Association of Early Childhood Development with Prevalence of Psychoactive Substance Use in Mexican Municipalities.
- Abstract: Abstract should be revised and make it more appealing. Include conclusions. They must be fit to the aim of the study. There are logical errors (see lines 22-23). Why provide data on homicide if this is not the aim of the study?
AUTHOR RESPONSE: Thank you for your comments. We edited the abstract We edited the abstract considering your suggestions to make it clearer. Regarding the data on homicide, we found that this factor has a strong association with ECD, although this is not the aim of the study we reported it in the results and addressed it in the discussion. The abstract now reads as follows:
Children’s early development is influenced by characteristics of the child, family, and environment, including exposure to substance abuse. The aim was to examine the association of early childhood development (ECD) with prevalence of psychoactive substance use in Mexican municipalities. We obtained ECD from the 2015 Survey of Boys, Girls, and Women (ENIM, for its Spanish acronym), measured with the ECD Index. The prevalence of psychoactive substance use was estimated at the municipal level, using the 2016 National Survey of Drug, Alcohol, and Tobacco Use (ENCODAT, for its Spanish acronym). Multilevel logistic models were fitted to evaluate the association between drug use and inadequacies in ECD overall and in four specific ECD domains: socio-emotional, physical, literacy--numeric, and learning. Inadequate ECD was directly associated with illegal drug use (OR = 1.10; 95% CI: 1.03, 1.17). For the specific ECD domains, inadequate socio-emotional development was directly associated with illegal drug use (OR = 1.08; 95% CI: 1.01, 1.15). These findings suggest that exposure to illegal drug use may be associated with ECD, and especially can lead to socio-emotional problems, although this cannot be considered the unanimous determinant of the problems presented. The implementation of evidence-based interventions to prevent drug abuse are necessary.
- Introduction: Please clarify the rationale of the study, provide examples of studies by other authors that meet the stated aim. A reference is needed for the NIDA study (lines 59-61).
AUTHOR RESPONSE: Thank you for your comments. We addressed your comments as best we could and provided some references for the NIDA study. The changes made in the introduction are as follows:
Existing literature on the topic suggests that the neighborhood environment has been identified as key for supporting families to cope with adversities and threats that could affect child development [1,4]. Exposure to drugs is one of the many disadvantageous neighborhood influences on child development. There is evidence that children living in negative environments have greater exposure to substances and more opportunities to use them [10].
All these investigations met in pointing out the impact of parental drug use on child development. However, to the best of our knowledge, no prior research has examined how the frequency of use of psychotropic substances in a municipality in which the child is growing up impact on ECD. This study has compiled information from various sources, in order to explore how the prevalence of psychoactive substance use in municipalities are associated with overall ECD and its specific domains in Mexico. The main hypothesis was that children with exposure to non-prescription use of medical drugs and illegal drug use could present language, motor, and cognitive deficits during growth. On the other hand, children living municipalities with the highest number of homicides in Mexico would have worse ECD outcomes.
Poner referencia NIDA
- Material and Methods: The assessment of substance use data is unclear (lines 114-119). Please describe this in more detail and provide references. How were the first and second level data combined? How many municipalities were included in the analysis?
AUTHOR RESPONSE: We edited the methods to explain the assessment of substance use data and how the first and second level data were combined, as follows:
We used a harmonized dataset of individual and municipal-level data for 145 municipalities. Based on the child’s place of residence, we linked individual-level with municipal-level data using unique municipality codes. Municipalities are second-level administrative divisions (states being the first). They have legislative and executive authority and are responsible for the provision of basic public services for their population.
2.2. Study Measurements
2.2.1. Psychoactive substance use
The prevalence of non-prescription use of medical drugs (opiates, tranquilizers, sedatives, and amphetamines) and of illegal drug use (marijuana, cocaine, crack, hallucinogens, inhalants, heroin, and methamphetamines) in the 12 months prior to the survey was calculated using ENCODAT data. Then the respondents were asked “Have you ever, even once, used marijuana, cocaine, crack, hallucinogens, inhalants, heroin or methamphetamines in the past 12 months?”, and “Have you ever, even once, used opiates, tranquilizers, sedatives or amphetamines in the past 12 months?”. Therefore, we considered two binary variables: non-prescription use of medical drugs (Yes/No) and illegal drug use (Yes/No).
We finally selected the Fay-Herriot estimator because it improved direct estimator in terms of accuracy. We used the R ecosystem to implement the SAE methodology, and a Fay-Herriot model for estimated proportions was adjusted with the function mseFH from the SAE library[24].
- Results: For OR, it is enough to use two decimals but this is not a principal requirement.
AUTHOR RESPONSE: Thank you, we have modified the decimals in the results section but we decided to use three decimal places in the tables.
- Discussion: Please discuss the main results of the study "the association between drug use and inadequate early childhood development (ECD), both overall and in four ECD domains." Is this a direct or indirect association? How can you explain the nature of this association? Compare with studies by other authors that have investigated such an association.
AUTHOR'S RESPONSE: Thank you, we have modified the discussion, adding some sections and improving the readability.
- Conclusions: There are presented only implications of the study but there are any conclusions generated in the present analyses.
AUTHOR RESPONSE: Thank you for your suggestion, we changed the text, it now reads:
To our knowledge, this research presents the first empirical evidence in Mexico of a link between psychoactive substance use and inadequate ECD. We provide evidence that exposure to illegal drugs are pervasive in the lives of children. However, although illegal use drug plays a role as an important risk factor for child development, this cannot be considered the unanimous determinant of the problems presented because other environmental factors that influence child development. These findings suggest that interventions and the implementation of public policy to prevent the use of psychoactive substances may benefit children's development, with significant benefits for social environment, and they should be an ongoing priority in the public health field. Child development is one of the United Nations Sustainable Development Goals [55], and governments should consider interventions to improve municipal conditions and to promote early childhood development, implementing new programs for prevention and treatment for drug use and disorders in the municipalities.
- References: References must be presented accordind to the requirements of the journal.
AUTHOR RESPONSE: Sorry for the confusion. The references have been fixed.
- English spell revision is necessary.
AUTHOR RESPONSE: Thank you very much for your comments. This manuscript was edited for proper English language.
